# Active Learning of Ordinal Embeddings:
# A User Study on Football Data

**Christoffer Löffler**[1,2]                                            *christoffer.loeffler@fau.de*
**Dario Zanca**[2]                                                       *dario.zanca@fau.de*
**Björn Eskofier**[2]                                                   *bjoern.eskofier@fau.de*
[2] *Friedrich-Alexander-Universität Erlangen-Nürnberg*
*Machine Learning and Data Analytics (MaD) Lab,*
*Carl-Thiersch-Straße 2b, 91052 Erlangen, Germany*

**Kion Fallah**[3]                                                       *kion@gatech.edu*
**Stefano Fenu**[4]                                                      *sfenu3@gatech.edu*
**Christopher Rozell**[3]                                                *crozell@gatech.edu*
[3] *School of Electrical and Computer Engineering*
[4] *School of Interactive Computing*
*Georgia Institute of Technology, Atlanta, GA*

**Christopher Mutschler**[1]                               *christopher.mutschler@iis.fraunhofer.de*
[1] *Precise Positioning and Analytics*
*Fraunhofer Institute for Integrated Circuits (IIS)*
*Nuremberg, Germany*

**Reviewed on OpenReview:** *https://openreview.net/forum?id=oq3tx5kinu*

## Abstract

Humans innately measure the distance between instances in an unlabeled dataset using an unknown similarity function. Distance metrics can only serve as a proxy for similarity in information retrieval of similar instances. Learning a good similarity function from human annotations improves the quality of retrievals. This work uses deep metric learning to learn these user-defined similarity functions from few annotations for a large football trajectory dataset. We adapt an entropy-based active learning method with recent work from triplet mining to collect easy-to-answer but still informative annotations from human participants and use them to train a deep convolutional network that generalizes to unseen samples. Our user study shows that our approach improves the quality of the information retrieval compared to a previous deep metric learning approach that relies on a Siamese network. Specifically, we shed light on the strengths and weaknesses of passive sampling heuristics and active learners alike by analyzing the participants' response efficacy. To this end, we collect accuracy, algorithmic time complexity, the participants' fatigue, time-to-response, qualitative self-assessment and statements, as well as the effects of mixed-expertise annotators and their consistency on model performance and transfer learning.

## 1 Introduction

Position tracking of persons, vehicles or objects is ubiquitous and enables various trajectory data mining tasks (Zheng, 2015), such as match or performance analysis in popular sports like football (Löffler et al., 2021), hockey (Chen et al., 2005) and basketball (Sha et al., 2016), or (public) transport planing and mobility analysis (Fernández et al., 2017; Shen et al., 2019; Yadamjav et al., 2020). With each additional moving agent, the data become increasingly complex due to its often unstructured and high-dimensional nature. For example, football has 23 trajectories (players and ball) that are typically tracked via multi-camera video systems Löffler et al. (2021). This affects tasks such as similarity-based information retrieval, which

queries similar occurrences to a given query scene (Sha et al., 2016). Here, a scene is an ensemble of agents' trajectories in a window of time.

In such large and complex datasets, information retrieval requires two expensive steps. First, the unstructured trajectories are optimally assigned, e.g., using the Hungarian algorithm Kuhn (1955). Second, the pairwise distance between the matched pairs of trajectories is computed on the raw trajectory data. Already by themselves, the two steps do not scale well to more realistic datasets that can have both a large number of samples and high dimensionality.

Recently, convolutional Siamese networks were leveraged by Löffler et al. (2021) to learn approximations of both the trajectory assignment and distance metric. The resulting lower-dimensional representation enables the scaling up of Euclidean distance-based information retrieval. However, there are two limitations. First, the Euclidean distance in itself is limiting, e.g., it does not weigh subjectively more important trajectories higher, and is affected by the data's high dimensionality (Canessa et al., 2020). Second, following directly from estimating the Euclidean distance of high dimensional data, the embedding captures global ordinal structures better than local ones (Löffler et al., 2021).

This work proposes to actively learn a distance function from human annotations. This learned similarity function is independent of the data's dimensionality and leads to a lower-dimensional ordinal embedding that more closely matches human perception. We address the associated costs of the human-in-the-loop by adapting an Active Learning (AL) sampler. We hypothesize that human annotations preserve more relevant information than the Euclidean distance and that a neural network can learn this perceived similarity from few annotations. We pay special attention to mixed-expertise annotators.

Our method learns rank-ordering from relative comparisons of a tuple of data instances using an active query selection method. In our experiments, we choose InfoTuple (Canal et al., 2020) over Crowd Kernel Learning (Tamuz et al., 2011) as it generalizes triplet queries to arbitrary tuple size and queries the oracle with more informative tuples. This has clear benefits: due to the tuples' larger size, they provide more context and are easier to annotate, and are also more sample efficient. We conduct a user study to evaluate the Active Learning sampling and use a well-suited football trajectory dataset for it. The study is centered around challenging relative comparison queries to human annotators. The query tuple composition impacts the efficiency of annotators, who may skip hard queries (see pre-study in Appendix A.1). Furthermore, while participants may have different similarity functions in mind, we can use a proxy similarity function for pre-training (Sha et al., 2016; Löffler et al., 2021). Crucially, the study allows a broader evaluation than accuracy per labeled sample. It also evaluates practically important questions such as the impact of skipped responses for different tuple composition algorithms, and it reports the annotators' intra-rater consistency and inter-rater reliability. We use a questionnaire for qualitative self-assessment of expertise and perceived similarity. These properties make our study suitable for the evaluation.

We pose three broader research questions. **RQ₁**: How consistent and reliable are the participants? **RQ₂**: What is the best baseline heuristic to generate tuple queries? **RQ₃**: Does active sampling perform better than non-active sampling? Our experiments then also answer which methods are the most suitable with respect to pure improvement in effective accuracy, relative time efficiency (i.e., user response time), or sampling efficiency (i.e., number of tuples skipped). We conclude with a study on how users' response consistency affects the learned metric and the capacity for transfer learning. We summarize our contributions as follows

- We reduce the computational complexity of the InfoTuple active learner.
- We experimentally analyze the efficiency of our method with a user study on active learning methods on real-world data, analyzing strengths and limitations.
- We answer the research questions and show the adapted InfoTuple active sampling leads to higher triplet accuracy than (non) active sampling methods but falls behind slightly in sampling efficiency. In addition, participants can form relatively consistent groups of user-specific similarity functions.
- We release a simple but effective web app for active learning experiments[1].

---

[1]Code available at `https://github.com/crispchris/Active-Learning-of-Ordinal-Embeddings`

The rest of this article is structured as follows. Section 2 formulates the problem and presents AL methods and adaptations. Section 3 explains our experimental design, including the procedure, metrics, participants, and dataset. We present the results in Section 4 and discuss them in Section 5. Section 6 concludes.

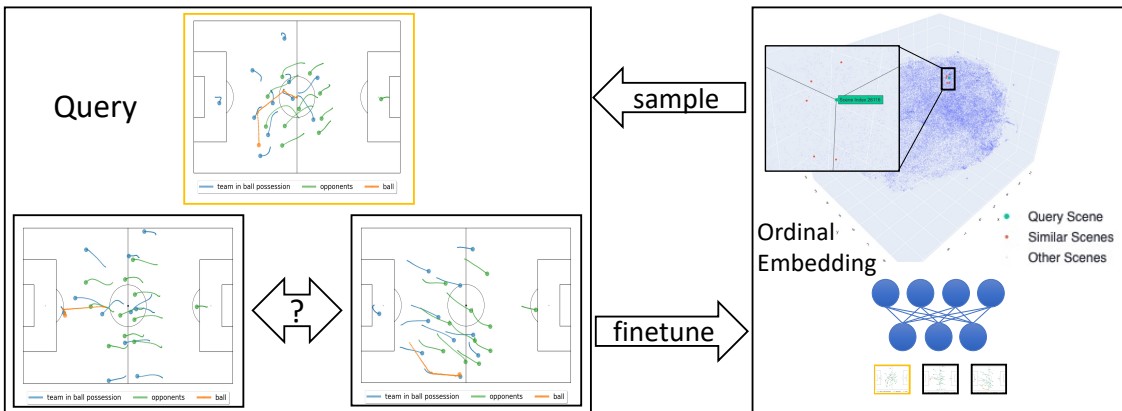

(a) Left: We collect participants' responses to relative similarity queries. In our study, we use a tuple size of nine. Right: We finetune a neural network to learn an ordinal embedding. We use different (active) sampling methods to generate queries. We show a 3D UMAP (McInnes et al., 2018) plot of the learned embedding purely for visualization.

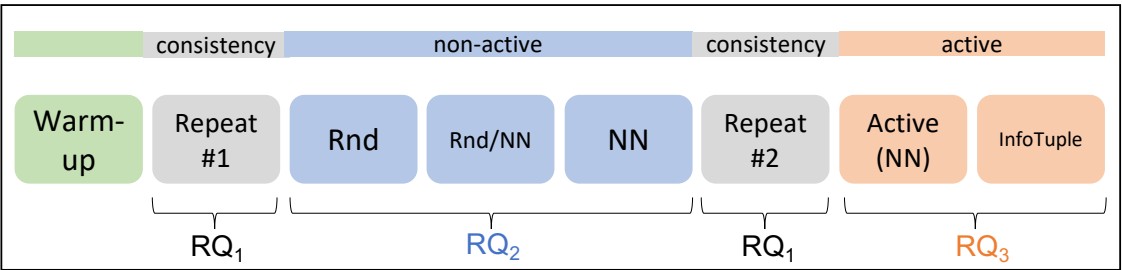

(b) We ask participants to annotate InfoTuples in sequential phases. We start with a warm-up, then measure consistency in two repeated phases ($\mathbf{RQ}_1$). The main research questions are answered in a non-active ($\mathbf{RQ}_2$) and in an active phase ($\mathbf{RQ}_3$). The query tuple composition heuristics in $\mathbf{RQ}_2$ are Random (Rnd), Nearest Neighbor (NN), and their combination (Rnd/NN).

Figure 1: The study consists of several different annotation phases. (a) shows the procedure to collect annotations and to finetune and sample from the learned embedding. (b) shows the multi-phase study design that compares different sampling strategies and evaluates model performance with respect to efficiency and effectiveness.

## 2 Method

Tuple composition and sample selection are important for learning an ordinal embedding specific to a human's similarity function. We formulate the problem first as pairwise similarity learning using a Siamese network like Löffler et al. (2021), which initially approximates assignments through metric learning (Section 2.1). This delivers a meaningful embedding to warm-start tuple selection strategies. Next, Section 2.2 extends the objective to triplet-based learning to generate an ordinal embedding. Section 2.3 then explains optimizations of InfoTuple, and our adaption to select the most informative samples from a meaningful candidate set.

### 2.1 Problem Formulation

We first investigate calculating pairwise distances of spatio-temporal matrices $\vec{x}$ of dimensionality $S \times T$:

$$\vec{x} = \begin{bmatrix} x_{1,1} & x_{1,2} & ... & x_{1,T} \\ x_{2,1} & x_{2,2} & ... & x_{2,T} \\ \vdots & \vdots & \ddots & \vdots \\ x_{S,1} & x_{S,2} & ... & x_{S,T} \end{bmatrix},$$

where the number of the dimensions of the trajectory $S$ is the spatial dimension ($S = 2$ for 2D positional tracking), and the number of time steps $T$ is the temporal dimension ($T = 125$ for 5 s tracking at 25 Hz). A trajectory may correspond to an agent like a football player. A row vector $\boldsymbol{x}_{s,1:T}$ represents a single dimension $s$ over time T. Positional tracking in sports may be calculated from multi-perspective video feeds Löffler et al. (2021). For 5 s scenes of one player sampled at 25 Hz, this produces trajectories $\vec{x}$ of the dimensionality $2 \times 125$.

In multi-agent tracking, a scene $\boldsymbol{X}$ consists of $N$ different trajectories such that $\boldsymbol{X} = \{\vec{x}^{(1)}, \vec{x}^{(2)}, .., \vec{x}^{(N)}\}$ (Löffler et al., 2021). See Fig. 1a for three scenes. Hence, $\boldsymbol{X}$ is of dimensionality $N \times S \times T$. For a scene with all 22 players and 1 ball, this results in a size of $23 \times 2 \times 125$. However, the ordering of the dimension $N$ is unknown, because agents may follow different strategies in each scene and can adapt their role in a game as needed Bialkowski et al. (2014). This is problematic for calculating the distance between matrices (Löffler et al., 2021). Specifically, given a pair of matrices $\boldsymbol{X}_1$ and $\boldsymbol{X}_2$, the respective pairwise assignment of vectors $\vec{x}^{(i)}$ and $\vec{x}'^{(j)}$ from each matrix is unknown.

Hence, to calculate the matrices' pairwise distance, we first compute the optimal row-wise assignment in the dimension $N$ of $\boldsymbol{X}_1$ and $\boldsymbol{X}_2$ using the Hungarian algorithm (Kuhn, 1955), that minimizes the sum of pairwise distances between the matrices in $\mathcal{O}(n^3)$. For this we calculate the distance $d$ between two trajectories $\vec{x}$ and $\vec{x}'$ as the average Euclidean distance over the spatial dimensions at each point in time (Löffler et al., 2021):

$$d(\vec{x}, \vec{x}') = \frac{1}{T} \sum_{t=1}^{T} ||\vec{x}_{:,t} - \vec{x}'_{:,t}||_2 \tag{1}$$

The distance between ensembles of trajectories $d_{ens}(\boldsymbol{X}_1, \boldsymbol{X}_2)$ is simply the sum of distances between trajectories (as in Eq.1) of matrices $\boldsymbol{X}_1$ and $\boldsymbol{X}_2$, that are optimally assigned along dimension $N$ (using the Hungarian algorithm). In terms of sports tracking data, this procedure maps players from one scene optimally onto players of another scene such that the distance between scenes is minimal.

We then further follow Löffler et al. (2021) and use Deep Siamese Metric Learning with a Temporal Convolutional Network (Lea et al., 2016) with a Resnet architecture (He et al., 2016) f to learn an approximate assignment and lower dimensional, distance-preserving embedding. The observations are pairs of matrices $\boldsymbol{X}_1$ and $\boldsymbol{X}_2$ sampled uniformly at random from all possible permutations of the dataset. Our goal is to find an embedding that preserves the distance $d_{ens}(\boldsymbol{X}_1, \boldsymbol{X}_2) \approx \hat{d}_{ens}(\widehat{\boldsymbol{X}}_1, \widehat{\boldsymbol{X}}_2)$ where $\hat{d}_{ens}$ is the Euclidean distance between the learned lower dimensional representations $f(\boldsymbol{X}_1) = \widehat{\boldsymbol{X}}_1$ and $f(\boldsymbol{X}_2) = \widehat{\boldsymbol{X}}_2$. Hence, the learning objective can be formulated as

$$\mathcal{L}(\boldsymbol{X}_1, \boldsymbol{X}_2) = \left( ||f(\boldsymbol{X}_1) - f(\boldsymbol{X}_2)||_2 - d_{ens}(\boldsymbol{X}_1, \boldsymbol{X}_2) \right)^2. \tag{2}$$

We use two $L_2$ regularization terms. The first $||f(\boldsymbol{X}_1)||_2 + ||f(\boldsymbol{X}_2)||_2$ centers learned representations and the second $||\theta||_2$ performs weight regularization on the network f's parameters $\theta$. $\hat{d}_{ens}$ is the Euclidean distance. It is simple to implement but still measures play similarity just as well as, e.g, Dynamic Time Warping, Fréchet distance or $l_\infty$ distance (Sha et al., 2016).

$\boldsymbol{X}_1$ and $\boldsymbol{X}_2$ contain trajectories of multiple *agents* which leads to a computationally expensive assignment problem (Sha et al., 2017) between two sets of trajectories describing two different scenes. This is because role

assignments are not fixed and often even disjoint between teams. Sha et al. (2017) build a tree structure to perform the assignment, that others subsequently use to dynamically set role-based assignments independent of a priori positions (Di et al., 2018). In contrast, we estimate these assignments similar to Löffler et al. (2021) (based on trajectory data and agnostic of roles) who compare three approaches: random assignments, exploiting additional meta-information such as roles, or inferring spatial proximity from data. The last variant uses a fixed grid template and assigns rows to channels, but this introduces sparsity in these inputs, which become overdetermined, as there are fewer players than entries in the grid template. Instead, we propose an improved template matching algorithm: we center a circular template with as many available positions as rows over the spatial center of $\boldsymbol{X}$. Then, we fit the template's variance in each dimension $S$ to match $\boldsymbol{X}$. For a pair $\boldsymbol{X}_1$ and $\boldsymbol{X}_2$, the template fitting and matching produces representations, with that we can estimate pairwise distances with lower error than Löffler et al. (2021), but without introducing sparsity in the network's inputs. See Appendix A.4 for the experimental comparison of the variants.

In summary, this approximates and reduces the computational complexity for $s$ scenes from $\mathcal{O}(s \cdot n^3)$ down to $\mathcal{O}(s \cdot m)$ where $m$ is the size of the network's embedding.

## 2.2 Active Metric Learning

Learning a metric from human annotators is costly. Active Learning helps reduce the amount of labeled relative comparisons by querying the oracle with informative samples (Settles, 2009; Houlsby et al., 2011). Active Learners use a model state, an acquisition function, and a query format.

**Initial model state**. Performing active sampling on a non-random model state is called warm-starting. Since there are no ground-truth labels available, we can use the learned Euclidean embedding to warm start AL methods, as inspired by Simo-Serra et al. (2015). The Euclidean embedding is independent of the participants' similarity function and can serve as a more general early estimate of similarity. Following that, we may then fine-tune this pre-trained model to an initial set of annotations before actively selecting queries.

**Acquisition function**. For learning from relative comparisons, the acquisition function needs to construct the query $Q$ from the most suitable candidate samples from the dataset. Following the notation from Canal et al. (2020), a tuplewise query $Q_n$ at time step $n$ of the AL procedure has a "head" object $a_n$ and an un-ordered "body" of objects $B^n = (b_1^n, b_2^n, ..., b_{k-1}^n)$. Now $Q_n = (a_n, B^n)$ denotes the $n^{\text{th}}$ tuple query, and a participant's ranking response $R(Q_n) = (R_1(Q_n), ..., R_{k-1}(Q_n))$ which is a permutation of $B^n$ rank-ordered by similarity such that $R_i(Q_n) \prec R_j(Q_n)$, if $i < j$. This indicates that the oracle ranks the object $R_i(Q_n)$ more similar to the anchor $a_n$ than the object $R_j(Q_n)$ (Canal et al., 2020).

Next, we follow a principled approach for an initial Active Learning heuristic. Given the head $\boldsymbol{X}_a$, we categorize closer samples in the Euclidean embedding as either positive $\boldsymbol{X}_p$ or negative $\boldsymbol{X}_n$. Xuan et al. (2020) further break down neighbors based on their distance to $\boldsymbol{X}_a$. Hard negatives are close to the head but dissimilar, whereas easy negatives are far away from it and least similar. Of these two types, the hard samples are most beneficial for learning, whereas easy ones produce no useful gradient (Xuan et al., 2020). Conversely, hard positives are highly similar samples, that are far away from $\boldsymbol{X}_a$, and thus mining these for tuples is difficult. Easy positives on the other hand are naturally available in an appropriately pre-trained embedding. Xuan et al. (2020) show that selecting easy positives keeps intra-class variance and helps to avoid over-clustering of the embedding.

We adapt the triplet mining concepts to Active Learning. Query tuples $Q_n$ composed of $\boldsymbol{X}_a$'s nearest neighbors can contain both easy positives and hard negatives. The other samples with higher distance from $\boldsymbol{X}_a$ are hard positives and easy negatives and are least useful when learning similarity. Nadagouda et al. (2022) recently proposed a similar NN acquisition function to collect both similarity and classification responses, see Sec. 2.4.

**Query format**. Generally, an ordinal embedding can be learned from tuples of size $s \geq 3$. A triplet loss (Chechik et al., 2010) over the three samples $\boldsymbol{X}_a, \boldsymbol{X}_p, \boldsymbol{X}_n$ is defined as follows:

$$\mathcal{L}(\boldsymbol{X}_a, \boldsymbol{X}_p, \boldsymbol{X}_n) = \max(\| \mathrm{f}(\boldsymbol{X}_a) - \mathrm{f}(\boldsymbol{X}_p)\|_2 - \| \mathrm{f}(\boldsymbol{X}_a) - \mathrm{f}(\boldsymbol{X}_n)\|_2 + 1, 0) \tag{3}$$

Canal et al. (2020) show that human annotators can benefit from larger tuple sizes than three. Query tuples $Q_n$ or arbitrary size greater than three can provide more context for considering the similarity and can be more sample-efficient. We can simply decompose a response $R(Q_n)$ into triplets $t = \{\boldsymbol{X}_a, \boldsymbol{X}_p, \boldsymbol{X}_n\}$ and use triplet loss. In this work, we determine the tuple size experimentally via a pre-study.

To summarize, we establish *Active (NN)* as an Active Learning heuristic, that constructs queries of arbitrary size from the nearest neighborhood of an anchor sample in an embedding.

### 2.3 Informative Queries

The selection of easy positives and hard negatives may not reliably construct the most informative queries. InfoTuple (Canal et al., 2020) improves upon this by maximizing the information gain of new queries. Given a probabilistic embedding, and previous and candidate queries, InfoTuple then selects that query that maximizes the mutual information that a response provides. It then uses a $d$-dimensional probabilistic embedding, such as t-Stochastic Triplet Embedding (tSTE) (Van Der Maaten & Weinberger, 2012) or probabilistic Multi-Dimensional Scaling (MDS) (Tamuz et al., 2011), to embed the dataset $\mathcal{X}$ of size $N$. Next, Canal et al. (2020) use the datasets's embedding $\boldsymbol{M} \in \mathbb{R}^{d \times N}$ to define a similarity matrix $\boldsymbol{K} = \boldsymbol{M}^T \boldsymbol{M}$. From this similarity, the authors calculate a $N \times N$ pairwise distance matrix $\boldsymbol{D}$ for $\mathcal{X}$. Given the queries $Q_1$, $Q_2$, ..., $Q_{n-1}$ and their responses $R(Q_1)$, $R(Q_2)$, ..., $R(Q_{n-1})$, they select the next query from a set of possible queries using the conditional entropy $H(\cdot|\cdot)$ of the next possible reply $R(Q_n)$:

$$\underset{Q_n}{\arg\min} \, H(R(Q_n)|R(Q_{n-1})) - H(R(Q_n)|\boldsymbol{K}, R(Q_{n-1})) \tag{4}$$

The equation trades of two terms: the first term selects for uncertain queries provided previous responses $R(Q_{n-1})$ while the second term selects for unambiguous responses $R(Q_n)$ that can be encoded in the similarity matrix $\boldsymbol{K}$ by responses $R(Q_{n-1})$. Balancing these two measures selects queries with high entropy, but that can be consistently annotated by the oracle.

Next, Canal et al. (2020) develop simplifying assumptions on the joint statistics of query and embedding to enable a tractable estimation via Monte Carlo sampling. Importantly, they assume a Gaussian distribution on inter-object distances, and only need to sample the distribution for the current query $Q_n$, as proposed by Lohaus et al. (2019). With these simplifications Eq. 5 and Eq. 6 depend only on the distance matrix $\boldsymbol{D}$:

$$H(R(Q_n)|R(Q_{n-1})) = H\left(\underset{D_{Q_n} \sim \mathcal{N}_{Q_n}^{n-1}}{\mathbb{E}}[p(R(Q_n)|\boldsymbol{D}_{Q_n})]\right) \tag{5}$$

$$H(R(Q_n)|K, R(Q_{n-1})) = \underset{D_{Q_n} \sim \mathcal{N}_{Q_n}^{n-1}}{\mathbb{E}}[H(p(R(Q_n)|\boldsymbol{D}_{Q_n}))] \tag{6}$$

They slightly abuse the notation by using $H(X) = H(p(X))$ for a probability mass function $p$ of random variable $X$, and $\mathcal{N}_{Q_n}^{n-1}$ to represent the Gaussian distribution $\mathcal{N}(D_{Q_n}^{n-1}, \sigma_{n-1}^2)$. We refer to Canal et al. (2020) for further details.

### 2.4 Adaption

In this work, we use a tandem of a generalizing neural network for learning similarity and InfoTuple's probabilistic model for selecting queries via Active Learning. We adapt InfoTuple to increase its effectiveness, leading to less ambiguous queries, improved sampling efficiency, and a reduction of the required time to select informative tuples.

**Effectiveness**. Following the assumptions in Sec. 2.2, we use the neural network's embedding to generate sample candidates to form the body $B$ from the neighborhood of a head $a$. This differs from the approach of Nadagouda et al. (2022), which selects the most informative nearest neighbor query from all possible queries. We do this because our problem domain suffers from sparse similarity. For instance, successful goals are

typically rare but passes in midfield are much more common, leading to a long-tailed, imbalanced distribution of sample similarity with sparsity in the tail (Wang et al., 2010). Thus, queries may often provide no positive and informative candidates. This leads to a high rate of skipped queries, which in turn leads to fatigue of the human annotators. We discuss this phenomenon in Sec. 5. This pre-selection from the Neural Network's embedding space increases the likelihood of finding similar candidates and reduces the subjective difficulty of the annotation task. Based upon a sampling of $m$ candidates, InfoTuple then selects the most informative query $Q_n$. The set of $m$ candidates is larger than the tuple size of $Q_n$ and is a hyperparameter (see Sec. 3.5).

**Efficiency**. The temporal complexity of the selection algorithm is primarily bound in the Monte Carlo style computation of Eq. 5 and Eq. 6. First, reducing the available candidate samples for greater effectiveness also reduces computational complexity. It leads to a smaller set of candidate samples available for constructing queries from. Second, we downsample the remaining $m$ samples like Canal et al. (2020), which further reduces the number of constructed queries $Q$ that have to be evaluated.

## 3 Experimental Design

In this section, we design a user study to answer the initial research questions.

### 3.1 Procedure

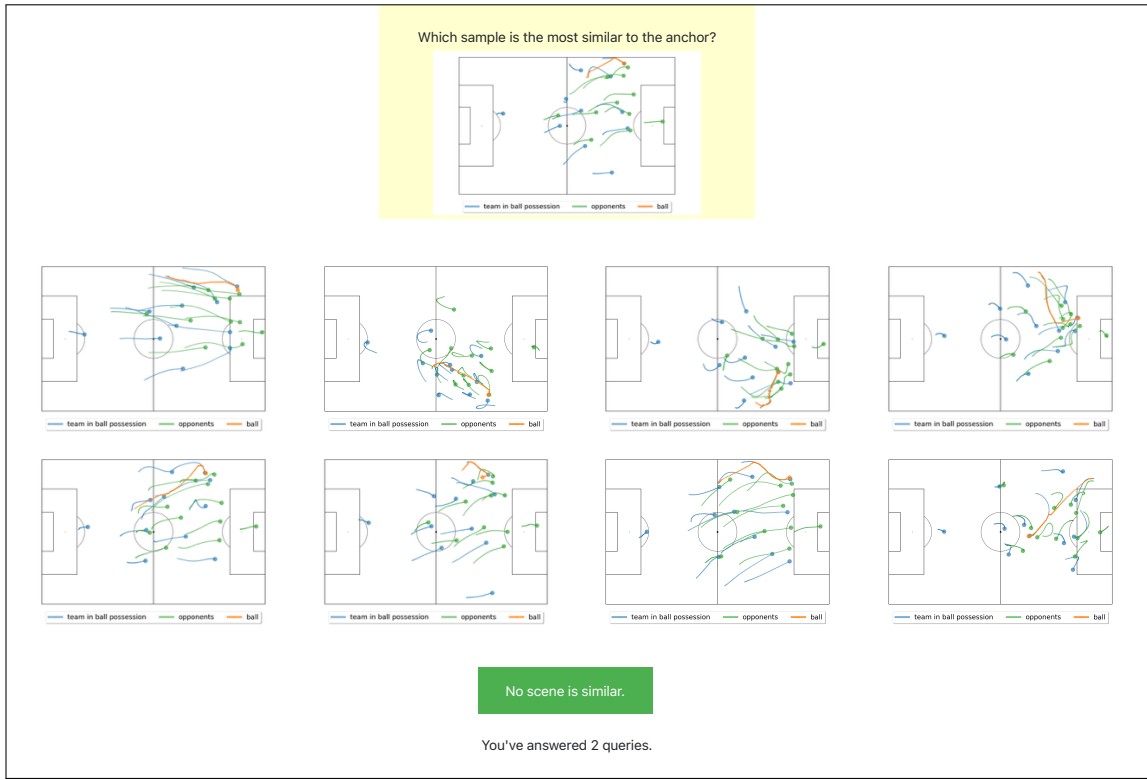

Figure 2: The web-based query page asks annotators to compare eight samples with an anchor and to either select the most similar sample or skip the query. Each sample shows the team in ball possession (blue), the defending team (green), and the ball (orange).

The study is designed in several phases and continuously collects annotations from 18 participants, see Fig. 1b. A smaller pre-study with 7 participants provided preliminary insights. With its help, we set the tuple size of InfoTuple to one anchor and eight samples for its body. The number of required samples for learning a meaningful ordinal embedding was about 250 to 300 triplets. Hence, the user study collects an equivalent number of tuples. Furthermore, we chose to train user-specific models, as the pre-study showed

that learned similarity is highly individual and does not generalize well. We include an evaluation of this notion in Sec. 4.4.

**Task description**. The task is to annotate an InfoTuple such as in Fig. 2. Participants compare an anchor with eight other samples, and then either choose the most similar sample, based on their own function of similarity or skip the query.

**Phases**. The user study is segmented into five parts. The first is an introduction with a description of the data and the mechanics of the annotation process. The second is a short warm-up of five queries, that familiarizes participants with the mechanics of the annotation website. Next, the split phase $\mathbf{RQ}_1$ occurs before and after $\mathbf{RQ}_2$. Finally, the active learning $\mathbf{RQ}_3$ concludes. $\mathbf{RQ}_1$ addresses intra-/inter-rater metrics. We repeat a fixed set of InfoTuple as Fig. 1b shows. We denote these sets as *repeated set* and highlight the phases as $\mathbf{RQ}_1$ in the figure. The repetition takes place after other phases, and the location of samples on the website is randomized, in order to break up any potentially lingering memories of the spatial layout of repeated queries. This way, we collect at least 20 InfoTuple but increase the set for each skipped sample. Hence, the repeated set may also grow in size as it also repeats skipped samples. Next, for the second research question, we collect annotations with different compositions of InfoTuple. The figure highlights the three segments as $\mathbf{RQ}_2$: *Rnd*, *Random/NN* and *NN*. We collect 20 InfoTuple to answer the research question and 10 more for testing. Finally, the last phase labeled $\mathbf{RQ}_3$ executes two different active learning strategies and trains neural networks with the annotations collected from participants. The first method is the Nearest Neighbor active learner, and the second is the adapted InfoTuple algorithm. We collect 20 InfoTuple with an acquisition size of one.

**Warm start**. We use a warm-start strategy for active learners. Starting from a pre-trained Siamese network, we finetune a model for each participant with their replies to the repeat #1 phase.

**Annotation tool**. The annotation tool is implemented in Python as a Flask web app. It uses the PyTorch framework for deep learning and integrates InfoTuple [2]. The user study was performed on an AMD Ryzen 7 3800X (64GB RAM) and an NVidia Geforce RTX 2070 Super (8GB VRAM), and participants used their clients to access a web frontend. Each participant had the same computing time and resources available.

## 3.2 Survey

We ask participants to self-assess in order to experiment with extensions from individual models toward generalized models of user-defined similarity. This serves as a qualitative, parallel approach to the intra- and inter-rater metrics.

We provide a simple questionnaire to inquire about the participants' self-assessment of their expertise and a qualitative description of their individual similarity function. We specifically ask them about their definition of similarity before and after the participation. This allows for qualitatively estimating the clustering of participants into groups that employ similar notions.

The questions were the following:

1. What is your expertise in football on a scale of 1 (novice) to 6 (expert)?

2. What will you look for when you compare football scenes' similarity?

3. Did your definition of similarity change?

4. What did you look for when you compare football scenes' similarity?

The first two questions were answered when looking at the tutorial. To capture any changes, the last two questions were answered after finishing all queries of the procedure. See Appendix A.2 for an analysis.

---

[2]https://github.com/siplab-gt/infotuple

### 3.3 Metrics

Triplet accuracy is defined as the agreement between two triplets $(a, b_1, b_2)$ and $(a', b'_1, b'_2)$ for a set of annotated triplets. In this study, we generate triplets from annotated tuples of arbitrary size.

Intra-rater consistency $C$ for a participant $P_i$ measures the agreement of annotations and skipped samples between the two phases Repeat #1 and Repeat #2:

$$C(P_i) = \frac{\text{ratings}_{\text{agreement}}(P_i)}{\text{ratings}_{\text{total}}(P_i)}. \tag{7}$$

Pairwise inter-rater reliability $R$ for two participants $P_i$ and $P_j$ includes numbers of all annotations and skips. It is the fraction of ratings in agreement over the total number of ratings:

$$R_{P_i, P_j} = \frac{\text{ratings}_{\text{agreement}}(P_i, P_j)}{\text{ratings}_{\text{total}}(P_i, P_j)}. \tag{8}$$

Response effectiveness (Bernard et al., 2018) $E$ is the measure of accuracy per time spend on a response. More effective sampling methods have a higher ratio relative to others.

$$E = \frac{\text{triplet accuracy}}{\text{response time}}. \tag{9}$$

We additionally evaluate the total effectiveness $TE$ of AL samplers, which includes the time spent sampling and training a model. While this depends on the choice of model and available resources for sampling and training, we consider it a relevant performance factor

$$TE = \frac{\text{triplet accuracy}}{\text{response time} + \text{computation time}}. \tag{10}$$

A high number of skipped responses may lead to participants' fatigue. Hence, we track the methods' label effectiveness $LE$. The accuracy relative to the number of skipped annotations provides a measure that extends the concept of efficiency (Bernard et al., 2018), which only measures the number of labeled instances over time.

$$LE = \frac{\text{triplet accuracy}}{\text{skipped responses}}. \tag{11}$$

### 3.4 Dataset

We use a dataset from the German Bundesliga from season 2014/15, that consists of trajectories sampled at $25\,\text{Hz}$ of 304 games, extracted from multi-perspective video feeds.

For our user study, we choose one game and extract scenes of $5\,\text{s}$ fixed length with $50\%$ overlap of $2.5\,\text{s}$. We further pre-process the data to simplify the implementation: we transform the data so that the team in ball possession players from left to right, we only use active (not paused) scenes, where one team has more ball possession than the other, and we require that no players are missing. This yields $1,005$ samples for the game. We consider the experiment size to be minimal but still representative. It allows us to control side effects so that we can see and compare the actual performance of the different methods. This helps to answer the research questions and still tests the proposed method.

We pre-train the baseline Siamese network similarly to Löffler et al. (2021). For this, we extract about 1,200,000 scenes from 304 games with a smaller overlap of $1\,\text{s}$, and use the same pre-processing. Then, we split the data into train/validation/test splits of $80/10/10\%$. As we cannot process all possible combinations of scene pairs, we instead randomly sample 10 million pairs for training and 1 million otherwise. Then, we train using the regularized Siamese loss (see Eq. 2) and the ellipse assignment method (see Sec. 2.1). We refer to Löffler et al. (2021) for additional details on the baseline's training and optimizer configuration.

**Test dataset**. There is no ground-truth dataset for user-specific similarity functions. A full set of comparisons would be unfeasible, given the combinatorial complexity, further motivating the use of Active Learning. Hence, for the test dataset, we collect additional hold-out sets of each participant's annotations. The inconsistency of annotations then impacts the possible accuracy score and introduces a baseline test error. Especially a low intra-rater consistency poses an upper limit of achievable performance.

Additionally, the composition of tuples itself may have a large impact on the test scores, as they represent different embedding neighborhoods. Annotations sampled at random for testing would provide information on the ordinal embedding's global order and are less biased. Test tuples from the anchor's neighborhood are based on the pre-trained Siamese embedding and thus are biased. However, they may provide some insights into the embedding's fine structure.

## 3.5 Experimental Setup

We use a Temporal Convolutional Network (Lea et al., 2016) with a Resnet architecture (He et al., 2016) like Bai et al. (2018) and Löffler et al. (2021). For its training, we use the triplet loss described earlier with an Adam optimizer with a learning rate of 0.001 and batch size of 32 for 10 epochs after each acquisition. We implement the experiments in PyTorch.

The InfoTuple method uses the tSTE to fit a probabilistic embedding. Furthermore, we select the hyperparameters for InfoTuple as follows. We sub-sample the candidates to 100 neighboring samples, that is about 10% of the whole dataset, and generate 10 random permutations as possible queries $Q_n$. We set the number of Monte Carlo passes to 10 and sub-sample the factorial of the 8-tuples with the factor 0.1.

# 4 Experiments

This section first presents the fundamental intra- and inter-rater metrics in Sec. 4.1. With this context, we give the sampling efficiency of the evaluated methods in Sec. 4.2 and we specifically show how the quality of annotators' replies affects the learning of their similarity functions in Sec. 4.2.1 Next, we analyze response times and the amounts of skipping in Sec. 4.3, before we conclude with a study on clustering of highly agreeing annotators in Sec. 4.4. We asked 18 participants (aged 20 to 35, one female) to take part in the user study.

## 4.1 Intra- and Inter-Rater Metrics

As a first step, which will allow us to better understand the results, we evaluate the annotators' intra-rater consistency and their inter-rater reliability. This allows us to analyze similarity functions based on the annotator's noise as well as discover possible groups of similarity functions.

**Intra-rater reliability**. We determine participants' reply consistency so that we can consider the noise level of their replies in later analysis.

The procedure contains two identical sets of queries (Repeat #1 and Repeat #2) that we compare to calculate consistency as defined in Eq. 7. Repeat #1 consists of at least 20 queries, with additional queries for each skipped reply. Repeat #2 then repeats the exact same queries per user. We shuffle the samples in the UI to avoid memory effects and repeat the set only after other annotation tasks.

Fig. 3a shows each participant's consistency from a low of 0.25 to a high of 0.75. The typical reply consistency clusters of 0.498 ($\pm$0.11), see Fig. 3b. We see two outliers, one to the low (p6) and to the high end (p11). The consistency agrees with the participants' self-assessment of their expertise as 1 for p6 and 6 for p11 on a scale of 1 (novice) to 6 (expert). Next, we normalize the self-assessment between 0 and 1, and show the difference to measured consistency in Fig. 3c. The self-assessed expertise typically matches consistency within 1 step up or down on the scale. This validates the quantitatively measured consistency. Furthermore, this score may help decide whether to collect annotations from a participant, or whether the generic baseline model is the better-performing alternative (see Sec. 4.2.1).

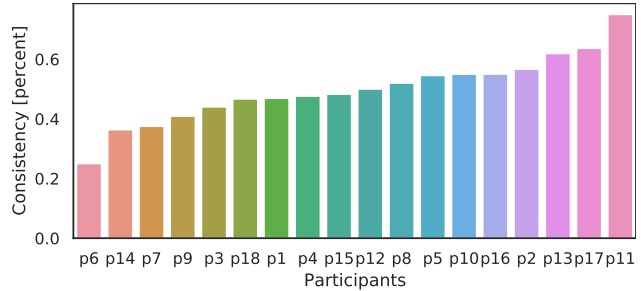

(a) Intra-rater consistency on repeated queries is from 0.25% to 0.75% (random baseline at 12.5%).

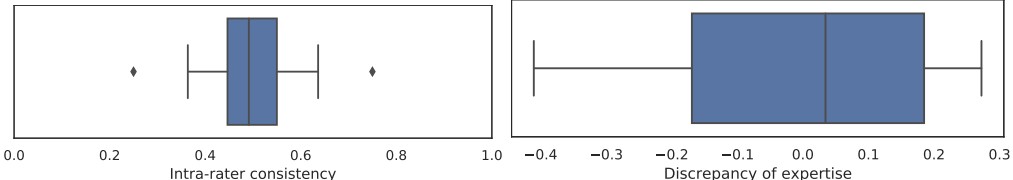

(b) The intra-rater consistency centers around 49.8% (±0.11) with two outliers at the extremes.

(c) Survey self-assessment minus their measured consistency shows the small difference between qualitative and quantitative expertise.

Figure 3: The intra-rater consistency in (a) shows how reliable participants determine similarity in a fixed and repeated set of queries, and we show their distribution in b. (c) shows that participants' self-assessed expertise in the survey differs from measured consistency by 20% or 1.2 points on the scale of 6.

These results show that the participants answer queries inconsistently, as the mean consistency to find the same similar samples from eight in total is $0.49(\pm 0.12)$. Additionally, the consistency measures identify unreliable oracles which help understand algorithmic performance better.

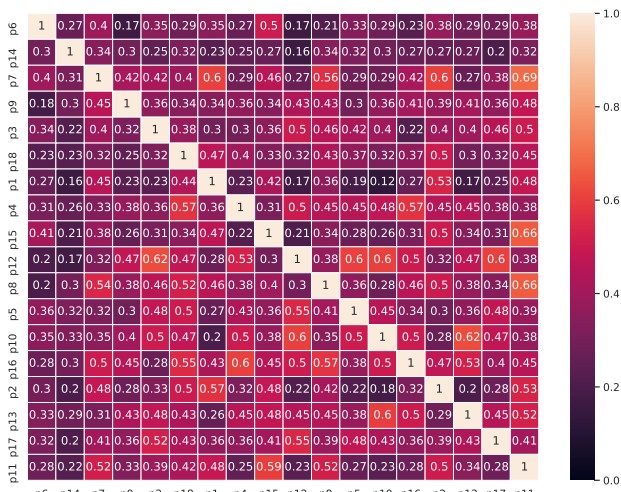

Figure 4: The inter-rater reliability on a fixed, repeated dataset shows whether participants form clusters of similarity functions, or whether they have orthogonal notions of similarity.

**Inter-rater reliability**. The subjective similarity functions may be part of clusters. Similar metrics could be learned from annotations if participants' concepts of similarity are close enough. The identification of such groups may be beneficial for analysis and fine-tuning.

The repeated sets of queries (Repeat #1) are identical for all participants. Only the actual number of replies may differ depending on skipped replies. Hence, it allows us to calculate the inter-rater reliability as

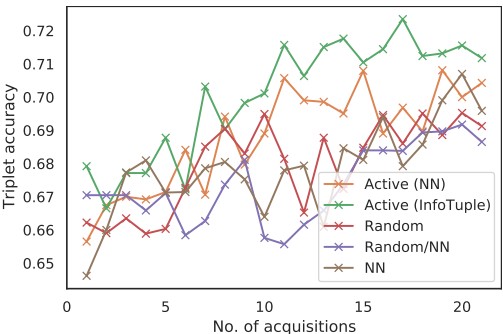

Figure 6: Triplet accuracy averaged over all users on the test set. We initialize the training data with the first *Fixed* dataset and then perform 20 acquisitions of size one. We train user-individual models and show the averages.

defined by Eq. 8. This way, we can identify sets of participants whose similarity functions overlap sufficiently to be considered as cluster members. We then use the reliability as the pairwise distance for hierarchical bottom-up clustering with maximum linkage.

We show a heat map of the pairwise reliability of the raters in Fig. 4 and a dendrogram of hierarchical clusters in Fig. 5. In the heatmap, we sort participants by their consistency from low to high. Higher inter-rater reliability is highlighted with brighter colors. Participants with lower consistency also show lower inter-rater reliability.

The intra- and inter-rater scores appear to be similarly ranged distributions. That means that the replies of some participants may be similar enough to cluster them together, allowing us to treat the entire cluster as one individual. The dendrogram shows hierarchical clusters of agreeing participants.

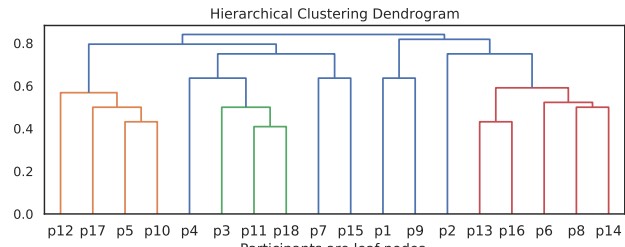

Figure 5: Hierarchial clustering dendrogram based on inter-rater reliability as pairwise distance.

We search for clusters of inter-rater reliability that have similar linkage as the intra-rater consistency, and highlight clusters with a linking distance equivalent to 0.37 or above, e.g., the four participants [p5, p10, p12, p17] form one such cluster.

We can determine clusters of similarity functions from these results. Still, the noise of replies can be in a similar magnitude as the reliability between participants, which corroborates the need for an analysis of transfer learning within such clusters. We will conduct this in Sec. 4.4.

## 4.2 Triplet Accuracy

The gains in predictive accuracy per annotated sample demonstrate whether a tuple composition is informative for the model. The accuracy represents the usefulness of the model for the application. We hypothesize that an active learning sampling of tuples is most efficient in increasing accuracy.

For this, we calculate the triplet accuracy on user-specific test datasets over 20 acquisition steps. For each participant, we first fine-tune the pre-trained network on 20 annotated InfoTuples from their Repeat #1 phases (equivalent to 140 triplets). We evaluate three non-active tuple composition methods as baselines (Rnd, Random/NN, NN) in comparison to two active learning methods (Active-NN, InfoTuple). The baselines are constructed from replies to fixed queries, the active learners were not fixed but interactively adapted to users. The test sets are user-specific sets of 10 annotated InfoTuple that were sampled randomly.

The test in Fig. 6 sees the active learners ahead, with InfoTuple on top. Purely randomly composed tuples compare well, especially in the first half of the acquisition phase, but level out around 69% triplet accuracy.

The remaining composition methods, which were constructed at least partially from the pre-trained network's embedding neighborhood stagnate for most of the duration of the experiment. These results for triplet accuracy are for 40 InfoTuples (280 triplets) in total. We further investigate the ceiling of the triplet accuracy, which is likely due to the noisy annotations and inconsistent replies, by training and testing user-individual models with all 120 InfoTuples (or 840 triplets) from the different annotation phases. The average triplet accuracy then reaches 73.7% ($\pm0.01$), which is only marginally higher than previously. The low gains with three times as many annotations show the diminishing returns of larger data collections.

Given the limited consistency of replies and the lack of ground truth data, there is an upper limit to the model's ability to learn. However, the relatively higher scores of InfoTuple are consistent and lead to increased model accuracy, whereas the NN active learner comes in second but similarly to the baselines. In terms of efficiency, InfoTuple is clearly preferable.

### 4.2.1 Noisy Oracles

Consistency of replies affects triplet accuracy and influences the network's ability to learn user-specific similarity functions. A lower consistency is harder to train with, and a pre-trained network performs better on test data than a fine-tuned one. More consistent annotations, however, enable fine-tuning. The InfoTuple active learner benefits from more consistent annotations.

We compare the triplet accuracy for the pre-trained model with all fine-tuned models for each user. We again use the users' self-annotated test sets *Rnd* and compare the accuracy. Additionally, we split the participants into groups of lower and higher consistency. For each group, we compare the pre-trained model with one trained using the InfoTuple active learner to test fine-tuning with more consistent annotations.

We show the effect of consistency on the InfoTuple algorithm in Fig. 7. For annotators with low consistency, the pre-trained network often performs similarly or even better than the finetuned variant. The pre-trained model is a strong baseline for inconsistent annotators. Fig. 7b shows that consistent annotations allow the active learner to learn a similar or better model than the pre-trained baseline, with 74% for the finetuned case vs 69.2% pre-trained on the test set. Appendix A.3 reports dis-aggregated triplet accuracy per user, supporting the results.

Generally, if the pre-trained network is already performing well on a test set, the fine-tuning with inconsistent data tends to decrease test triplet accuracy. It depends on the consistency of the similarity function used for annotations if fine-tuning is beneficial. For consistent participants, that also tend to be self-assessed experts, better user-specific similarity functions can be learned.

Furthermore, with InfoTuple as the AL sampler, we see strong increases in triplet accuracy over the pre-trained baseline (see Fig. 7). The most consistent $\frac{2}{3}$ participants lead to more stable learning and higher triplet accuracy, which is likely due to their more consistent replies or clearer notion of similarity (see Fig. 7b).

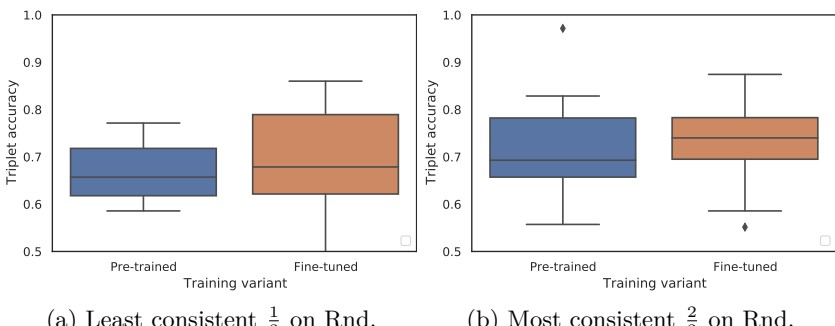

(a) Least consistent $\frac{1}{3}$ on Rnd.  (b) Most consistent $\frac{2}{3}$ on Rnd.

Figure 7: This figure shows the effectiveness of fine-tuning via InfoTuple active learning in comparison to the pre-trained baseline for different levels of consistency. Participants are grouped according to their consistency, into a least consistent third and the most consistent two-thirds. Relative gains are greater for more consistent replies.

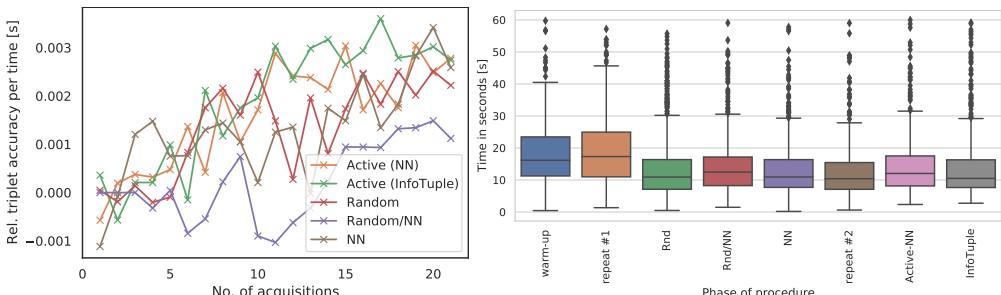

(a) Response Effectiveness over the number of acquisitions.

(b) The response times per phase, here as box plots over all users, show a relatively stable mean around 10 s for each phase of the procedure.

Figure 8: We present the triplet accuracy after applying Eq. 9 in (a) and show response times in (b).

### 4.3 Considering Time and Sampling Efficiency

We consider three different perspectives that shed some light on the relationship between triplet accuracy, time, and skipping. These metrics are relevant for determining the efficiency of the time spent, as well as the potential fatigue of participants.

#### 4.3.1 Response Effectiveness

The relative return in accuracy on the invested amount of response time is a relevant criterion when choosing a sampling method. The time that users spend analyzing samples and forming decisions may also be indicative of how difficult a query is to reply to. This excludes the time that methods require to propose queries.

We evaluate the response time for all users by dividing the relative gains in predictive accuracy by the participant's average response time. This yields a relative improvement in accuracy per time spend on a response.

We normalize triplet accuracy using Eq. 9 and report the results in Fig. 9. There, we see the highest increase in triplet accuracy for InfoTuple, followed by the NN active learner. The Random baseline follows closely. Interestingly, the strongest method InfoTuple is also the one that is replied to quickest. The response times of the 5 methods in our study are: Random with 9.7 s, Random/NN with 10.98 s, NN with 10.35 s, and the active samplers NN with 11.19 s and InfoTuple with 9.48 s, see Fig. 8b for an overview and Fig 14 in Appendix A.3 for user-specific details.

The relative return in accuracy should not be bought with exorbitant amounts of time. Our results show that both active learners perform comparably well due to their higher relative gains in accuracy compared to the baselines, which speaks for them as tuple composition methods.

#### 4.3.2 Total Effectiveness

While the response time itself is important for choosing a suitable sampling method, the overall time spent may also include an active method's computational overhead. The analysis is similar to Sec. 4.3.1 but also includes the time of methods to compute queries on top.

The algorithm execution times cause minor decreases in total effectiveness for the two Active Learning methods. The Nearest Neighbor active learner computes for 2.8 s ($\pm$ 0.59 s) and the InfoTuple active learner needs 6.11 s ($\pm$ 0.56 s). However, this depends mainly on the computational power and implementation efficiency. In this work, we optimized InfoTuple to decrease its complexity while still providing benefits over others.

Computation, such as the training of the neural network or the calculations performed by InfoTuple, impacts the accuracy-per-time efficiency. However, while we consider the cognitive load for participants to be taxing,

the waiting time between queries may be less demanding. In our user study, the additional waiting time of the active methods was still reported as *tough* by several participants and should be considered when selecting a method.

### 4.3.3 Label Effectiveness

We consider the number of skips during the labeling process as a potential source of participants' fatigue, because it takes longer to reach the set number of annotations. Hence, participants are queried repeatedly. Higher label effectiveness is to be preferred, see Eq. 11.

We present the least, the median, and the most consistent participants' timeline of skipped and annotated queries in Fig. 9 to highlight the prolonging effect, that a large number of skipped queries has on the duration of the experiment, and thus also on the participants' fatigue. In addition, we use Eq. 11 to calculate the label effectiveness of the different sampling methods. We report the details in Fig. 14 in Appendix A.3 and discuss the findings here.

The NN active learner combines an apparently easier-to-answer query composition with the benefits of active learning and increases accuracy (by the number of skipped queries) with $LE = 1.95\%$, followed by non-active with $LE = 1.39\%$. InfoTuple follows closely with $LE = 1.26\%$ due to its raw gains of accuracy. Even though more queries were skipped, responses were more informative overall. Random and Random/NN sampling score worst due to the high amount of skips or lower gains in accuracy with $LE = 0.75\%$ and $LE = 0.35\%$ respectively. Furthermore, we may infer that skipping unsuitable queries may be necessary to annotate more consistency, see $p6$ with $p15$ or $p11$ in Fig. 9. The mean percentage of skipped samples is 22.22% for NN queries, closely followed by Random/NN sampling (22.36%) and the NN active learner (24.53%). Both methods compose queries from close neighbors of the query anchor, i.e., likely similar samples. InfoTuple queries are skipped 50.1% of the time and random tuples form the rear with 53.97%.

The methods that primarily focus on NN sampling perform best due to the higher likelihood of positive samples in a query. InfoTuple samples from a larger pool of most informative samples, which may cause participants to skip more queries. However, its queries tend to be more informative so that InfoTuple keeps up with the other methods.

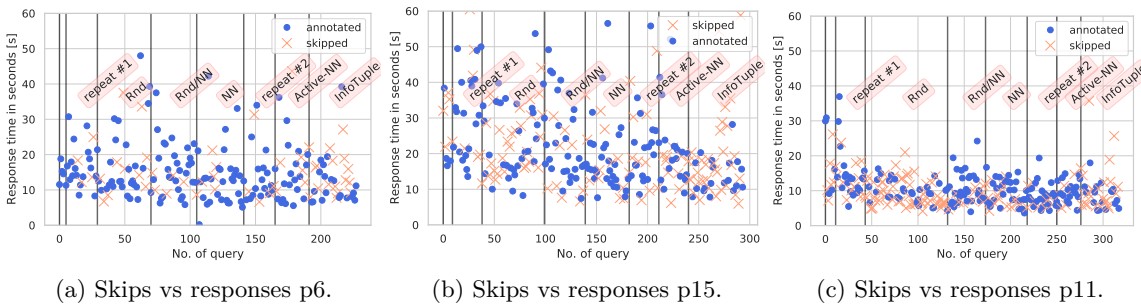

(a) Skips vs responses p6.     (b) Skips vs responses p15.     (c) Skips vs responses p11.

Figure 9: All response times for the least, the median, and the most consistent participant. We highlight the annotation phases.

### 4.4 Data Augmentation with Annotations

Participants' similarity functions may align well enough to profit from combining their annotations in order to augment their training dataset. This would increase sampling efficiency, as a larger collection of annotations would be available for warm-starting the training. Due to the comparable levels of intra-rater consistency and inter-rater reliability, such *clusters* seem realistic.

Our analysis selects the pair of participants with the highest inter-rater reliability. We exclude skipped samples and we select the pair with more consistent participants. we then initialize the training dataset with the combined set of both participants' Repeat #1 replies, fine-tune one model for each participant on their annotations selected by InfoTuple, and test on the participants' own test sets. Recall that since the queries,

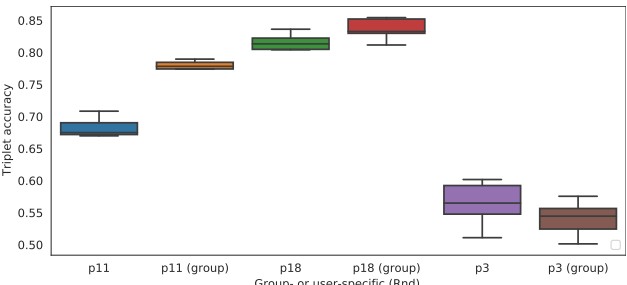

Figure 10: The effectiveness of grouping the training data from participants with similar consistency and notions of similarity. It can be beneficial to combine their warm-up training datasets to increase triplet accuracy.

that make up the Repeat #1 data are identical, the combined annotations augment the training with replies to skipped queries and can even contain different replies.

We select the participants $p3, p11$, and $p18$ according to the hierarchical clustering in Fig. 5, because $p11$ is the most consistent participant overall. The results in Fig. 10 show the respective participant's differences on the test set. Two of the participants benefit from additional annotations, the third sees a further decrease in accuracy.

Augmenting the warm-start dataset with the annotations of similar annotators can be beneficial. Our results for a triplet of participants, which also includes the highest inter-rater reliability, confirm it. This suggests that similarity functions may be clustered and used for transfer learning. However, our additional experiments with larger clusters yield diminishing returns and confirm our pre-study's findings, likely due to lower inter-rater reliability.

## 4.5 Discussion

**RQ**$_1$: How consistent and reliable are the participants?
Our evaluation shows that the participants are able to respond to the queries with relatively high quality, as their intra-rater consistency is in the range of 0.498 ($\pm0.11$) on average. Their self-assessment in the range of 1 to 6 is relatively consistent with a divergence of 20%, which we show indicates whether a user-specific data collection is beneficial. Furthermore, we found that noisy oracles affect the network's ability to learn user-specific similarity functions. However, even though the more consistent group benefited more from fine-tuning, the tendency does also points towards a smaller increase for less consistent annotators.

**RQ**$_2$: What is the best baseline heuristic to generate tuple queries?
The composition of tuple queries has implications beyond the learned model's triplet accuracy. Participants may take longer to respond or even find no response at all, which ultimately leads to fatigue. We show in Fig. 6 that the heuristics Random and NN yield a very similar triplet accuracy with Random/NN as a close third. However, once we account for the average response time, the gap between NN and Random on the one hand, and Random/NN on the other increases. We see two possible explanations. First, the Nearest Neighbor sampling generates queries from the relatively sparse Euclidean embedding, that are more likely to contain easy positives that participants select more quickly. Conversely, queries generated by the Random heuristic contain easy negative samples and are often quickly skipped, see Fig. 9. The skipping renders Random sampling less efficient, as the procedure's sampling phases require more queries to collect the same number of annotations, which leads to higher fatigue. Finally, the combination of Random/NN mixes easy positives and negatives and seems to require closer inspection by participants and is less sample efficient because of this, see Fig. 8b.

Overall, the NN heuristic balances the three metrics triplet accuracy, time spent on responses, and the number of skips and fatiguing of participants better than the other baselines Random and Random/NN.

**RQ$_3$**: Does an active sampling method perform better than baselines?

We show that annotations can be more valuable due to Active Learning, as the differences between ActiveNN and the baseline NN heuristics show. Furthermore, selecting informative samples with InfoTuple increases sample efficiency compared to ActiveNN, because a purely NN-query may not be maximally informative due to the sparse similarity of our problem domain. Response times are comparable to the heuristic baselines. Interestingly, the increases in accuracy per skipped sample are in favor of the ActiveNN sampler compared to InfoTuple, while the absolute triplet accuracy of models trained using InfoTuple is higher. Additionally, we show that the gained accuracy per time justifies the computational complexity of AL. Overall, Active Learning and specifically InfoTuple perform better than heuristic baselines.

**Quality of Annotations**

The quality of annotations is an important factor in machine learning. Specifically for Active Learning, applications typically deal with expensive domain experts, that may perform more reliably, but whose time is more expensive. Alternatively, non-expert annotators may seem more cost-effective but are likely to be less reliable (Wu et al., 2020). In our study, we address the scenario of mixed expertise and evaluate the deep metric learning objective with respect to quantitative and self-assessment. We show that self-assessment can be predictive of the usefulness of Active Learning compared to a strong baseline metric learner. We show that transfer learning can be applied successfully using more consistent, and potentially also semantically similar annotations.

**Broader Applicability of the Proposed Method**

The ideas proposed in this paper are twofold. First, the proposed deep active learning method may be applied to other tasks. Learning similarity of trajectory data is a common problem, e.g., in other team sports like basketball (Sha et al., 2016), hockey (Chen et al., 2005) or Australian football (Alexander et al., 2022), in animal farming (Chen et al., 2005), or in personal mobility, traffic or public transportation (Shen et al., 2019; Yadamjav et al., 2020; Fernández et al., 2017). Still, the propose method uses a flexible learner that may be adapted for other data than trajectories, e.g., learning similarity in image datasets such as Food73 (Wilber et al.) or ranking the age of faces (Liu et al.) and learning similarity between texts (Neculoiu et al.). Generally, the proposed method may be considered with the aim to train (or fine-tune) a generalizing Metric Learner from few relative comparisons of large tuple size with the help of a human-in-the-loop.

**Broader Evaluation of Active Learning**

The second idea that we propose is our call for a broader evaluation of new active learning algorithms. In Active Learning, the predominant performance metric seems to be accuracy per labeled sample. There are, however, additional practically important considerations besides accuracy. Experimental platforms, such as NEXT (Jamieson et al., 2015), capture detailed statistics on, e.g., network response time, timing data that helps to estimate human fatigue, and also the quality of annotations, i.e., via intra-rater agreement. More recent AL studies Canal et al. (2020) additionally analyze the tasks' difficulty and response time. Hence, in our work, we do not only rank methods by accuracy. Our analysis of effective accuracy over (total) time spent on responding, and the efficiency of accuracy by skipped queries support the results. The adapted InfoTuple method, which evaluates the Nearest Neighbors sampled from the neural network embedding, shows to be less fatiguing as well.

## 5 Related Work

Many approaches for deep metric learning use triplets to learn generalizing representations (Hoffer & Ailon, 2015; Xuan et al., 2020). Recently, Xuan et al. (2020) introduced the idea of sampling triplets with easy positive examples. This addresses the issue of over-clustering of the learned embedding, and it is tolerant of high-class variance. This is highly relevant to our study because it motivates the composition of tuple queries. Our study evaluates the effect of easy positive and negative samples with a human-in-the-loop.

Active learning uses insights into the data distribution or model views to select the sample(s) that best improves the model's fit of the underlying data distribution. We can group approaches for deep learning into different categories. Some estimate model uncertainty to select the most informative (Houlsby et al., 2011; Kirsch et al., 2019) or uncertain samples (Gal et al., 2017; Beluch et al., 2018), others select representative samples using a covering set (Sener & Savarese, 2018) or combine diverse and uncertain selections (Ash

et al., 2020). Nadagouda et al. (2022) use information theory to perform active metric learning and active classification on a deep probabilistic model that uses Monte Carlo dropout sampling (Gal & Ghahramani, 2016) as an approximation of uncertainty. They sample the most informative NN queries from all possible ones. Similarly, Canal et al. (2020) learn an ordinal embedding by maximizing the mutual information of queries to an oracle. Their work is based on a probabilistic model, fitted to relative comparisons, that learns the ordinal embedding. Both Nadagouda et al. (2022) and Canal et al. (2020) use larger tuple sizes than three, which improves sample efficiency and also provides more context for easier annotations.

Information retrieval from multi-agent trajectory datasets has to first solve the assignment problem of these agents' trajectories (Sha et al., 2017; 2016; Löffler et al., 2020), and second, provide a quickly searchable representation (Sha et al., 2016; 2017; Di et al., 2018; Löffler et al., 2020). Di et al. (2018) additionally learn ranking from humans. They estimate the assignment of agents (players) within a tree-structure (Sha et al., 2017) and then train a convolutional autoencoder on 2D trajectory plots to extract features. In contrast, we train one neural network Löffler et al. (2020), that learns to solve the assignment problem and produces a lower dimensional embedding, such that additional tree structures are redundant and search is accelerated. Di et al. (2018) learn ranking from participants of a click-through study, that collects only pairwise comparisons, and trains a linear rankSVM on these tuples and the extracted features of the autoencoder. This approach differs from this paper, as we optimize the network's embedding directly. However, the embedding may be used for on-top learning to rank approaches (Di et al., 2018) or even active embedding search (Canal et al., 2019).

# 6  Conclusion

In this work, we studied how to learn unknown similarity functions, that humans innately use to measure similarity between instances in an unlabeled, unstructured dataset. To learn these metrics from few annotations, we adapted mining easy positive triplets from a query sample's neighborhood in a neural network's embedding space and applied InfoGain to construct the most meaningful queries from the sparse similarity of a football trajectory dataset. Our user study shows the benefits of this method and provides a nuanced evaluation with respect to accuracy and the practical considerations of the effectiveness of Active Learning with a human-in-the-loop: response, total and label effectiveness. An accompanying survey sheds light on the mixed expertise of participants, their diverse notions of similarity, and their relation to a strong metric learning baseline. Future work may leverage the user-specific ordinal embedding to perform an iterative active search, that can further improve information retrieval.

**Author Contributions**

The main author Christoffer Löffler implemented algorithms, designed and conducted experiments, wrote and revised the manuscript. Kion Fallah and Stefano Fenu contributed to the theoretical framework and experimental design, offering insights. Kion Fallah and Dario Zanca provided manuscript feedback. Christopher Rozell guided the research, including experimental and algorithmic design. Christopher Mutschler supervised and helped refine the experiments and methodology, and provided manuscript feedback. Björn Eskofier supervised the project and provided manuscript feedback.

**Acknowledgments**

We would like to acknowledge support for this project from the IFI programme of the German Academic Exchange Service (DAAD), the Bavarian Ministry of Economic Affairs, Infrastructure, Energy and Technology as part of the Bavarian project Leistungszentrum Elektroniksysteme (LZE) and the Center for Analytics-Data-Applications (ADA-Center) within the framework of "BAYERN DIGITAL II". We thank the study participants for their valuable contributions. Special thanks to Nicolas Witt and Robert Marzilger for their support in organizing the study.

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

# A   Appendix

## A.1   Pre-Study

We conducted a pre-study to set the size of InfoTuple and also the number of required annotations for user-individual model training. We furthermore have discovered that a model would not generalize well from the complete, diverse group to one hold-out user. We also collected feedback on the introduction (tutorial) and query format, which helped to improve it and to avoid confusion among participants.

**Size of InfoTuple**. The pre-study was conducted with 7 participants, that were split into two groups *A* with 4 participants and *B* with 3 participants. Group A was queried with tuples of size 7 (1 head, 6 body), and group B with smaller tuples of size 4 (1 head, 3 body). We monitored response times and found that both groups responded within a similar amount of time. Furthermore, participants of group B reported that they often could not select a similar sample. Hence, we determined that a larger tuple size would not negatively impact response times, but would provide additional context that would help respond to the query, consistent with Canal et al. Canal et al. (2020).

**Count of InfoTuple**. Second, we set the required number of annotated triplets by, first, collecting 500 annotated triplets per participant, and second, fine-tuning a model for each in increasing sub-sets sizes, that

| group | notion | example |
|---|---|---|
| semantic | team movement | direction (attack or defense); dynamic or static |
| | ball movement | pass or shot; short or long; at rest |
| | semantic category | standards (kickoff); situations (pressuring, passing, shot); consistency of category |
| | team formation | compact; shifting; offensive or defensive |
| | area | which third; which flank |
| proximity | player and ball | Are there players near the ball? |
| | directions | direction of ball trajectory |
| | length of trajectory | |
| | distances | spatial proximity similar in candidate scenes |

Table 1: Qualitative clustering of responses to questionnaire into *semantic* and *proximity*.

we randomly sampled from the total pool. The increase in accuracy converged over the increasing number of training data. Hence, we chose to collect 20 InfoTuple of size 8 (140 triplets) per method, as more showed diminishing returns.

**Individual or general model**. Lastly, we observed that we were not able to train one model by combining the training data from all participants into one large pool. We used leave-one-out cross-validation to train on all but one participant and then tested on the remaining participant as a hold-out test dataset. The *general* model did not achieve as good results as training an individual model. Hence, we designed the user study such that individual models were trained.

### A.2 Notions of Similarity

We hypothesize that different notions of similarity may either already be captured by the Euclidean distance metric, or can be learned better from annotations. Recall that we have collected additional metadata on participants' notion of similarity using a questionnaire, see Sec. 3.2. This may help distinguish participant groups.

We test this hypothesis by analyzing the participants' responses to the questionnaire and extract two sets of rules, that we categorize as either *semantic* or *proximity*. Semantic rules describe the scenes' semantics whereas proximity rules largely ignore such interpretations. They instead pay closer attention to proximity or only to subsets of trajectories. To reduce bias, we ask two raters to assign participants to either group and then have the raters resolve their disagreements through an analytical discussion. Here, we compare the triplet accuracy of the agreed-upon group compositions.

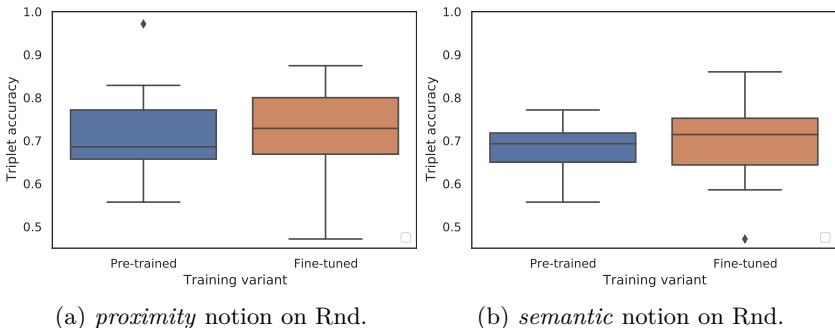

(a) *proximity* notion on Rnd.    (b) *semantic* notion on Rnd.

Figure 11: This figure distinguishes two types of similarity and shows the effectiveness of fine-tuning via InfoTuple active learning in comparison to the pre-trained baseline. Participants are grouped by their qualitative description of similarity, that is either proximity or semantic.

We show the extracted groups in Table 1 and each group's triplet accuracy in Fig. 11. We assign participants to either group if they describe a specific notion or provide examples that match it. The resulting *proximity*

group contains 8 participants and the *semantic* group 8, too. We excluded a participant that did not describe their notion and another that did not fit into any. We observe that high self-assessed expertise correlates with an assignment to the semantic group. Furthermore, we see an increase in triplet accuracy from pre-trained models to fine-tuned models for both groups in Fig. 11.

The increase in triplet accuracy is visible for both the semantic and the proximity groups. Hence, we can not show a difference from qualitative clustering of notions of similarity. However, fine-tuning can outperform the Siamese baseline in both cases, because the Euclidean similarity metric does not adequately capture either group's similarity function. In future work, alternative interpretations of qualitative and quantitative data may find a more suitable (sub) group segmentation.

### A.3 Additional Results

Fig. 12 shows the effectiveness of fine-tuning compared to the Siamese baseline per participant. Fig. 13 shows every participant's response times for each phase. Finally, Fig. 14 shows the adapted triplet accuracy for three variants of effectiveness: response, total and label effectiveness.

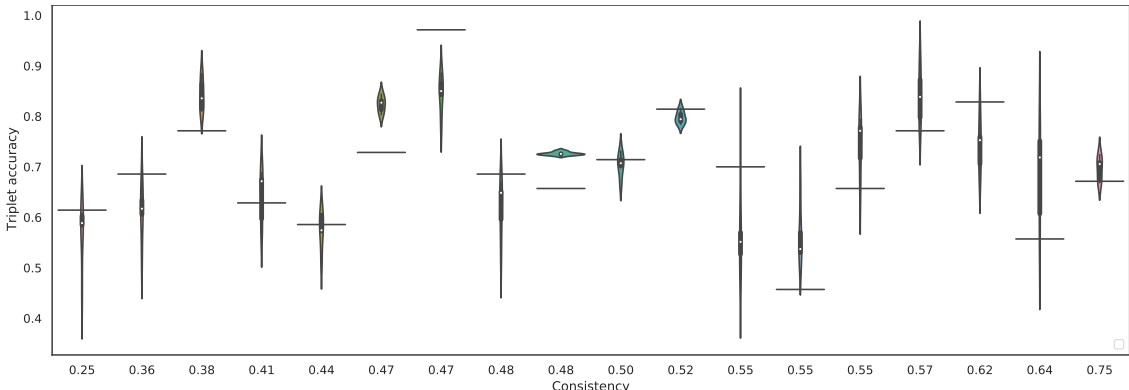

Figure 12: The effectiveness of all fine-tuning variants (violin plot) in comparison to the pre-trained Siamese network (horizontal bar) shows the impact of inconsistent oracles and a performance ceiling for the relatively noisy train and test data.

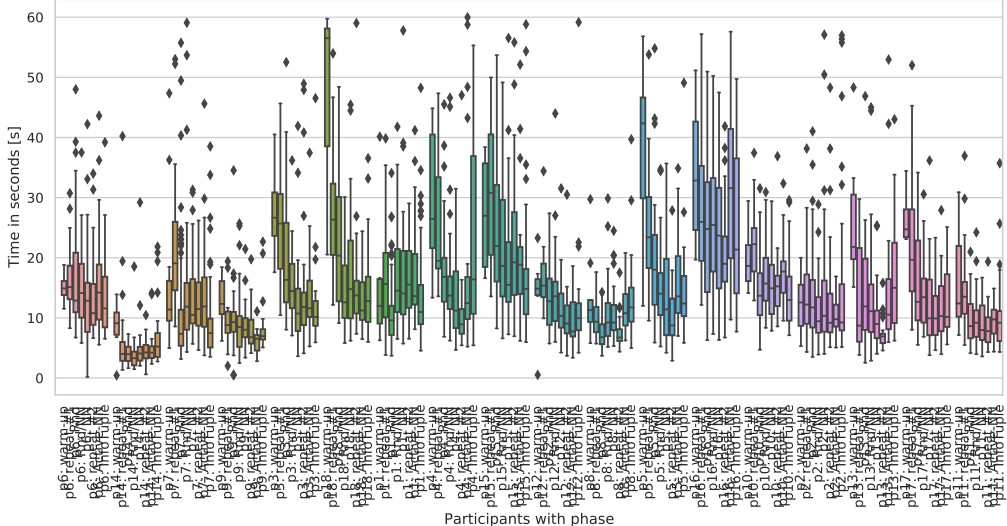

Figure 13: We show all users' response times for each phase ordered by consistency.

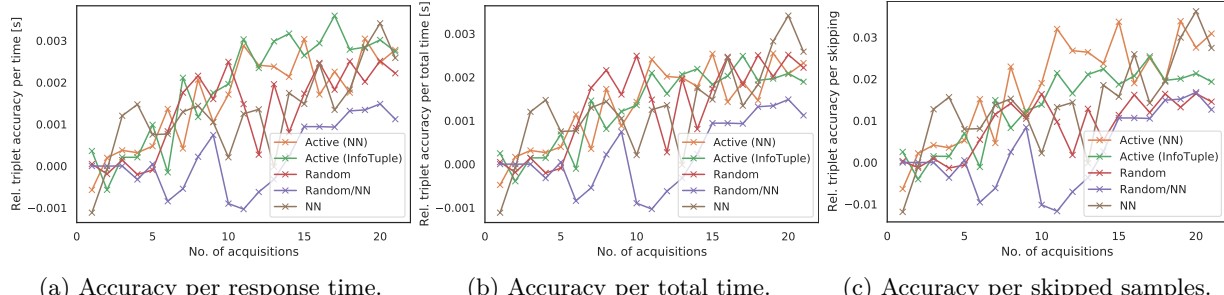

(a) Accuracy per response time.   (b) Accuracy per total time.   (c) Accuracy per skipped samples.

Figure 14: We average the results for all users on the test dataset. (a) shows the triplet accuracy per response time spent. (b) shows the triplet accuracy per time including compute time, which depends on compute resources and may limit the experiment's scale. (c) shows the triplet accuracy over the number of skips, which can cause fatigue.

## A.4 Evaluation of Ellipse Template

We present the experimental comparison of our proposed ellipse template matching algorithm from Sec. 2 with the assignment methods proposed by Löffler et al. (2021). Their best-performing method uses an over-determined role position assignment, that introduces sparsity in the inputs of the Siamese network because there are more positions in the template than trajectories in the scene. Our proposed ellipse template has the same number of positions as there are trajectories, and we fit the template to each scene, see Sec. 2 for details.

We reproduce the results of Löffler et al. (2021) and compare them with our proposed template matching algorithm. For that, we use a Siamese network with an embedding size of 64, pre-train it on 304 games (see Section 3.4), and perform an evaluation of ours and their method on the same subset of the dataset that we use in our main study. We follow their evaluation scheme and calculate the structural correspondence between the original data and the learned representations. They define the Mean Absolute Percentage Error (MAPE) as the expected absolute error relative to the ground truth, as

$$\text{MAPE}_{\mathcal{D}} = \frac{1}{|\mathcal{D}|} \sum_{i=1}^{|\mathcal{D}|} |\frac{d_{ens}(\boldsymbol{X}_{i,1}, \boldsymbol{X}_{i,2}) - \hat{d}_{ens}(f(\boldsymbol{X}_{i,1}), f(\boldsymbol{X}_{i,2}))}{d_{ens}(\boldsymbol{X}_{i,1}, \boldsymbol{X}_{i,2})}|. \tag{12}$$

where they sample and compare pairs of matrices $\boldsymbol{X}_{i,1}$ and $\boldsymbol{X}_{i,2}$ from a dataset of matrix pairs $\mathcal{D}$. We adapt our notation from Sec. 2 slightly, such that $\boldsymbol{X}_{i,1}$ refers to the first matrix of the $i$th pair and $\boldsymbol{X}_{i,2}$ to the second matrix. Likewise, $f(\boldsymbol{X}_{i,1})$ refers to the learned representation of $\boldsymbol{X}_{i,1}$.

For their best performing method role position, Löffler et al. (2021) report a MAPE of 2.66% on their validation and 2.68% on their test set. Paired with the embedding size of 64, this also achieves speedy retrieval due to the lower dimensionality. We reproduce a similar MAPE of 2.669% on our dataset, which is in line with their results. In comparison, our ellipse template achieved a MAPE of 2.31% with the same hyperparameters. This clearly shows the benefits of our proposed method, as its error is 0.359% lower, and at the same time, it eliminates input sparsity.

## A.5 Instructions

The user study uses a web-based annotation tool, that begins with a landing page with a short introduction. Participants are first presented with an introduction that familiarizes them with the data, the task, and the mechanics of the annotation tool. Fig. 15 shows the explanation of the data and description of details. Fig. 16 then describes the participant's task in this study. Finally, Fig. 17 shows an example of a query and once more explains the mechanics of the annotation process. After this participants begin annotating the warm-up samples with a button press.

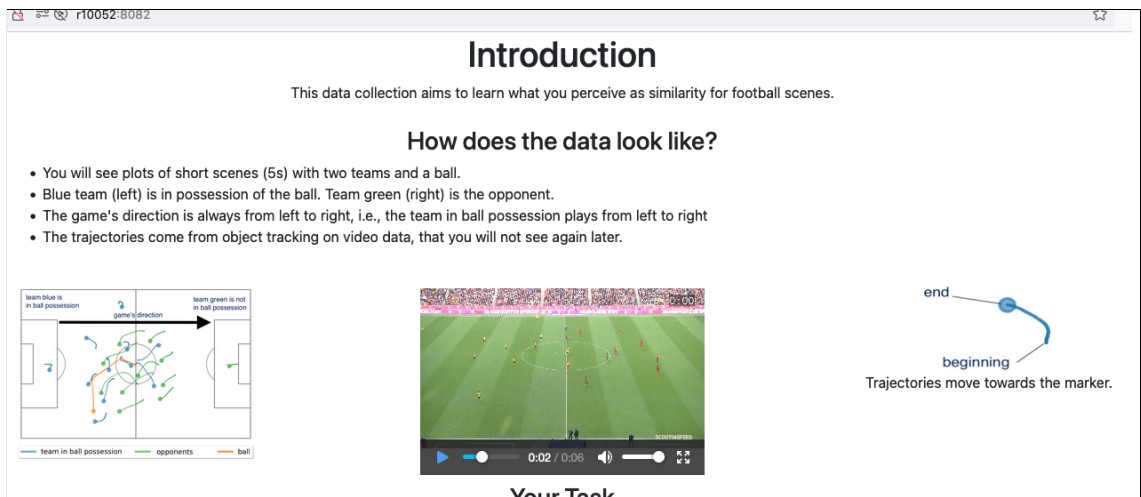

Figure 15: The landing page begins with an introduction to the dataset.

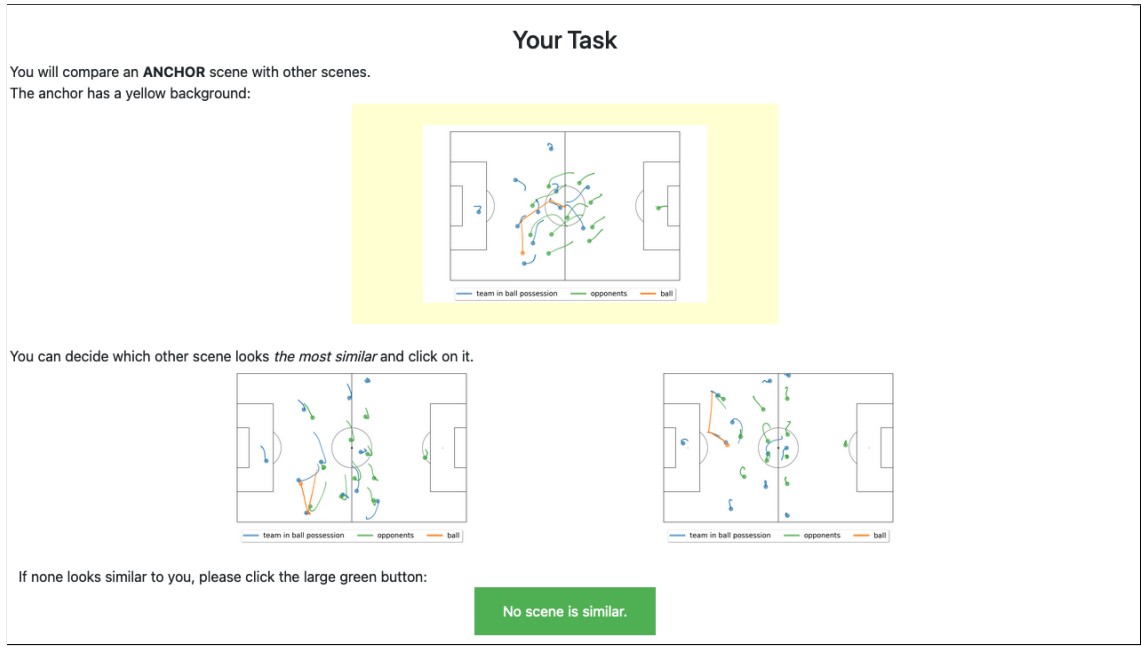

Figure 16: The next part describes the task and familiarizes participants with their options.

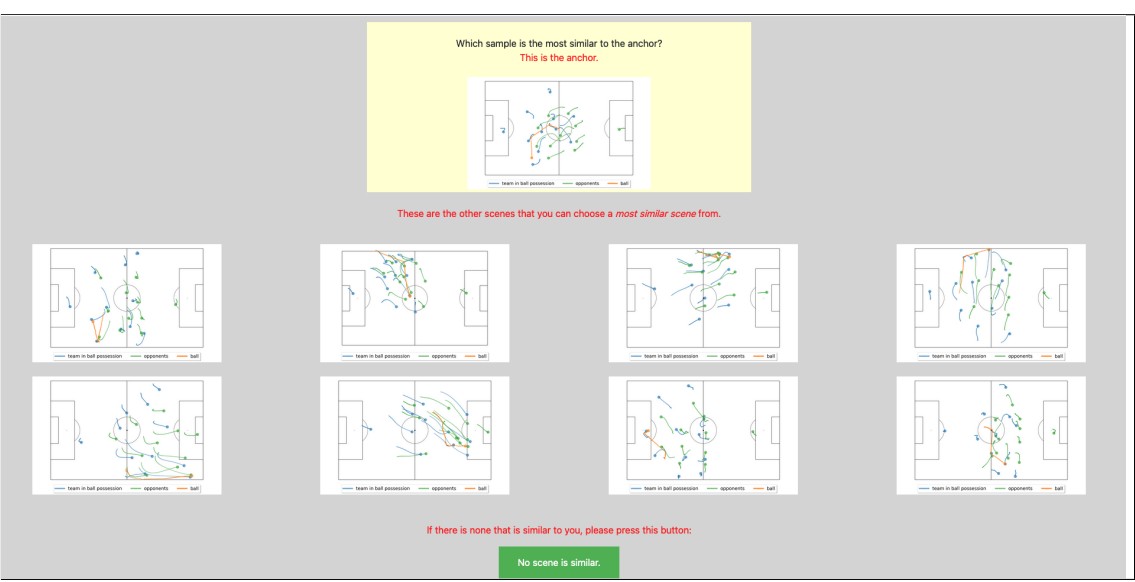

Figure 17: An example concludes the explanation and previews the annotation task that follows.

