# OpenReview forum: "Active Learning of Ordinal Embeddings: A User Study on Football Data"
_TMLR — Accepted by TMLR_

### Review · Reviewer_zZvF · 2022-09-05

**Summary Of Contributions:**

This work addresses the problem of learning user-defined similarity functions from few annotations. To this end, an entropy-based active learning method was employed to obtain annotations from human participants and use them to train a convolutional neural network (CNN).
The experimental results show that the proposed method improves the quality of the information retrieval compared to a baseline metric learning method. The strengths and weaknesses of passive sampling heuristics and active learners are examined in experimental analyses.

**Broader Impact Concerns:**

A broader evaluation of methods was given and some concerns were discussed.

**Requested Changes:**

Three major improvements are required:
1. Proof-reading paper by fixing mathematical notation and statements.
2. Either providing mathematical justification of claims, or revising the claims.
3. Improvement of the analyses using additional ablation studies with different deep neural networks.

**Strengths And Weaknesses:**

The paper is well written in general. Authors address an important problem with an interesting application. The research questions posed in the paper also highlight several practically important challenges. The proposed methods and questions were examined in experimental analyses.

Main weaknesses of the paper are as follows:

1. Mathematical statements and notation: Some notation such as the definition of B^n has typo. It can be also incorrect since you apply set union on B^n while it is defined as a list. There are several unclear notation and statements like this.

2. There are some over-claims such as “We prove the efficiency of our method with a user study on active learning methods on real-world data, analyzing strengths and limitations.”. Indeed, this claim is experimentally analyzed.

3. Following (2) and similar claims, experimental analyses of the paper should be improved. In addition, mathematical justification of the claims should be provided.

4. In the analyses, some simple CNNs have been employed. The results should be examined using different types of deep neural networks, or an ablation study with different CNNs should be provided to explore generalization of the results for different models.

---

> ### Author Response · Authors · 2023-01-02
> **Response to Reviewer zZvF**
>
> Thank you for your detailed review of our submission. We would like to address your requests. Please also take note of the revision with changes highlighted in red.
>
> **1. Proof-reading paper by fixing mathematical notation and statements.**
>
> We have followed your advice and revised the mathematical notation and statements in the sections "problem formulation" (Section 2.1), "active metric learning" (Section 2.2) and "informative queries" (Section 2.3). We fixed the mistakes and enhanced the detail of explanations of our problem formulation to make the sections easier to follow.
>
> **2. Either providing mathematical justification of claims, or revising the claims. For example "We prove the efficiency of our method with a user study on active learning methods on real-world data, analyzing strengths and limitations.”.**
>
> We carefully re-evaluated our claims. Regarding the claim on our study, that you have highlighted in the introduction (Section 1), we agree that we do not prove the efficiency but instead experimentally analyze it. Consequently, we have revised the statement.
>
> We have also expanded the evaluation of the claimed improved performance of our proposed template matching algorithm (problem formulation, Section 2.1) and added additional experimental results (Evaluation of Ellipse Template, Appendix 4). We are not aware of other unjustified claims. Do you agree with the revised paper's claims?
>
> **3. Improvement of the analyses using additional ablation studies with different deep neural networks.**
>
> The pre-trained neural network jointly learned to estimate the assignment problem and the (proxy) Euclidean distance metric (Löffler et al., 2021). Experimenting with transfer learning of another type of deep neural network means solving these two learning tasks again. For example, developing a novel embedding approach based on a recurrent network is out of scope for our user study. This paper instead proposes to actively learn a new metric function from human annotators. For estimating a proxy metric we would kindly refer to Löffler et al. (2021).
>
> **In the analyses, some simple CNNs have been employed.**
>
> We do not use a simple CNN in our experiments, but a TCN (Lea et al., 2016) with a Resnet architecture (He et al., 2016) that is considered state-of-the-art for sequence modeling (Bai et al., 2018). It is specifically developed for unordered ensembles of trajectories (Löffler et al., 2021). We revised the "problem formulation" (Section 2.1) to clarify the choice of network architecture, and state it again in the "experimental setup" (Section 3.5).

---

### Review · Reviewer_wkRi · 2022-11-29

**Summary Of Contributions:**

This paper describes a pipeline of active learning for the task of football trajectories matching. It describes a compreshensive description on how to design this system, which includes learning a convolutional network, sampling hard examples for human annotators (i.e. active learning), and other important details such as active learning. The submission also presents a detailed description of actual human study and shows that the proposed method performs better than other baselines such as popular prior method of matching the trajectories by eucledian distance or non-actively learning. Overall, I think the submission is technically solid.

The downside is that the prescribed methods are very much specific to the application of trajectory matching. Also, perhaps given my unfamiliarity with the application, I find the current draft hard to follow at times.

**Broader Impact Concerns:**

I do not see negative societal impact from this paper.

**Requested Changes:**

My main request is around writing. I think in the first few paragraphs of the intro, the paper should concisely define the task of "multi-agent position tracking". The paper should also motivate why this is an interesting and important task. Then in section 2.1, the paper should give concrete examples to the technical notations. For example, what is a "trajectory"? How is the data in X collected?

I also encourage the author to think about the general implication of their proposed method. Can some of the ideas here be applicable to other tasks as well?

**Strengths And Weaknesses:**

Strengths:
- empirically solid and provides a detailed description of the proposed system
- have abundant results to support the proposed methods and conducted real human study

Weakness:
- proposed method is very specific to the task and unclear if how the submission will be relevant to the general audience of TMLR
- The writing is not super clear for audience who are not already very familiar with the task of multi-agent trajectory tracking

---

> ### Author Response · Authors · 2023-01-02
> **Response to Reviewer wkRi**
>
> Thank you for your detailed review and helpful suggestions. We have revised the paper according to your requested changes (highlighted in red).
>
> **My main request is around writing. I think in the first few paragraphs of the intro, the paper should concisely define the task of "multi-agent position tracking". The paper should also motivate why this is an interesting and important task.**
>
> The revised introduction begins with a  motivation. We cite several applications in sports and transportation and refer interested readers to related work. Next, we define multi-agent position tracking with an example from sports. A new technical definition follows in "problem statement" (Section 2.1). We think that these changes motivate, introduce and explain multi-agent position tracking better than before.
>
> **Then in Section 2.1, the paper should give concrete examples to the technical notations. For example, what is a "trajectory"? How is the data in X collected?**
>
> We have revised Section 2.1 with concrete examples from football and a more detailed explanation of the terms "trajectory" and "ensemble of trajectories". The revision also uses more rigorous mathematical notation.
>
> We similarly revised the explanations of "active learning" (Section 2.2) and "informative queries" (Section 2.3) to clarify them further.
>
> **I also encourage the author to think about the general implication of their proposed method. Can some of the ideas here be applicable to other tasks as well?**
>
> Thank you for this question. We added the discussion of the broader applicability of the proposed method to the "discussion" (Section 4.5). We believe that the Deep Active Learning method of an ordinal embedding may be applied to trajectory mining in sports and also to other domains, such as transportation and mobility, farming, etc. It may also be adapted to other data types (images, text, etc.) with minor changes. For example, it would be interesting to apply it to image-based learning of similarity of food taste. This opens up an avenue for interesting future work.

---

### Review · Reviewer_1z6h · 2022-12-29

**Summary Of Contributions:**

This paper adapts InfoTuple for active learning of ordinal embeddings on football data. The main contribution is the application of InfoTuple on football data.

**Requested Changes:**

1. The dataset used is very small and one one dataset is used. The authors might need to conduct experiments on more larger datasets
2. The authors should also include more state-of-the-art network embedding approaches as baselines

**Strengths And Weaknesses:**

Strengths:
- This paper studies an interesting problem of ordinal embedding on football data
- The authors conduct extensive experiments

Weakness:
- The technical novelty of the paper is rather limited. It simply adopts InfoTuple for football data. The authors might want to submit to some application oriented journal.
- The dataset used is very small and one one dataset is used. The authors might need to conduct experiments on more larger datasets
- The authors should also include more state-of-the-art network embedding approaches as baselines

---

> ### Author Response · Authors · 2023-01-02
> **Response to Reviewer 1z6h**
>
> Thank you for your review of our submission. We would like to discuss your requests and have revised our submission accordingly (highlighted in red).
>
> **The technical novelty of the paper is rather limited. It simply adopts InfoTuple for football data. The authors might want to submit to some application oriented journal.**
>
> We believe that our work is within the scope of the journal for multiple reasons. The main contribution of our paper is that we adapt and extend InfoTuple with concepts from triplet mining to increase the efficiency by sampling nearest neighbors for query construction (see Sections 2.2 and 2.4). We conduct a real-world user study to test whether the model works. We use special metrics, e.g., "response effectiveness" and consider effects like fatiguing of participants (see Section 3.3), to measure more detailed real-world performance besides pure accuracy. Furthermore, we use a deep convolutional network to generalize to unseen samples, which is a different use of InfoTuple.
>
> For these reasons our work goes beyond an application of an existing technique, that sheds light on the strengths and weaknesses of the method, as the journal's submission guidelines state; it is also an extension of the existing method.
>
> **The dataset used is very small [...].**
>
> We found that we have not included the complete experiment description in the draft. We first pre-train the Siamese network on 304 games, and then perform the active learning user study on a subset of the data.
>
> We agree that larger experiments with more samples and study participants can be preferable in general. But we capped the dataset size of the user study, because we consider one game to be a minimal but representative sample size. This way, we control the experiment's side effects and costs, so that we can see and compare the performance of the different methods.
>
> We have revised the description of the "dataset" (Section 3.4) accordingly.
>
> **[..] only one dataset is used.**
>
> We extend the method for data with sparse similarity, e.g., we sample from the embedding neighborhood according to triplet mining literature (Xuan et al., 2020), see Sections 2.2 and 2.4). We consider the method applicable to a wide range of real-world tasks, e.g., different domains of trajectory mining applications (sports, transportation, animal farming, etc.) and data types (image, text, et cetera), see the revised discussion in Section 4.5 ("Broader Applicability of the Proposed Method"). Even though we could apply the method to other datasets, we did not conduct these additional user studies due to the focused aim of this paper and cost constraints.
>
> The contributions of this paper are centered around a real-world user study on learning human's innate similarity functions. For this purpose the constrained parameters of the experiment were enough to show the strengths and weaknesses of the extended method, which is our main contribution (see Section 4.5, "discussion"). However, in future projects on active learning of similarity metrics, we will consider performing additional user studies.
>
> **The authors should also include more state-of-the-art network embedding approaches as baselines**
>
> We use a Temporal Convolutional Network (Lea et al., 2016) with a Resnet architecture (He et al., 2016), that is considered state-of-the-art for sequence modeling (Bai et al., 2018). It is specifically developed for unordered ensembles of trajectories (Löffler et al., 2021).
>
> The pre-trained baseline neural network jointly learns to estimate the assignment problem and the (proxy) Euclidean distance metric (Löffler et al., 2021). Including another neural network embedding, e.g., based on a recurrent architecture, would imply developing a new approach. This is an interesting question for further research, but we consider it to be out of the scope of this submission. This is because our paper instead proposes to actively learn a new metric function from human annotators.

---

### Review · Reviewer_UmLB · 2023-01-08

**Summary Of Contributions:**

This work presents an active learning method to learn a distance function from few human annotations. In particular, the authors emphasize the tuple composition, sample selection, and a real case on the football dataset. Experiments are done to verify the proposed method.

**Broader Impact Concerns:**

None.

**Requested Changes:**

1. Careful proofreading
2. Clear Clarification on notations
3. Analysis of overparameterization risk on the small dataset or Empirical results on larger datasets

**Strengths And Weaknesses:**

Strength:
- The motivation for active learning of ordinal embeddings is reasonable and interesting.
 - A user study on football data is performed to verify the proposed method.

Weakness:
- Introduction describes less about the user study on football data. It would be better to motivate why this user study is suitable for the active learning method.
- Some notations are unclear or wordy. For example, in Equation 1, there seems no need to expand all elements in X; In Equation 2, what is the difference between $X_2$ and $X’_2$? I cannot find the definition of $X’_2$.
- The dataset is small. Any overparameterization risk? It means that the number of learnable parameters is much larger than the number of training samples. The authors should analyze the influence of this risk.
- There are some typos. For example, on page 3, “multiple agents which leads to” should be “… which lead to”.

---

> ### Author Response · Authors · 2023-01-09
> **Response to Reviewer UmLB**
>
> Thank you for your review of our paper. We have revised our submission and would like to address your comments. Please take note of the changes in the revision, highlighted in red.
>
> **Introduction describes less about the user study on football data. It would be better to motivate why this user study is suitable for the active learning method.**
>
> We have revised the introduction to include our motivation for conducting this user study as a suitable evaluation (Section 1). We also discuss the broader evaluation of active learning in Section 4.5.
>
> **Careful proofreading: There are some typos. For example, on page 3, “multiple agents which leads to” should be “… which lead to”.**
>
> We have proofread the draft and corrected the mistakes with the help of a more advanced spell checker. We have not highlighted these changes in the revision but we hope all of them have been addressed.
>
> **Clear Clarification on notations: Some notations are unclear or wordy. For example, in Equation 1, there seems no need to expand all elements in X; In Equation 2, what is the difference between $X_2$ and $X'_2$? I cannot find the definition of $X'_2$.**
>
> We have revised the mathematical notation and statements in Section 2.1, including Equations 1 and 2. $X'_2$ was actually a mistake that we have corrected. The revision fixes further mistakes and enhances the detail of explanations of our problem formulation. Please also see the revised Sections "active metric learning" (Section 2.2) and "informative queries" (Section 2.3). We think that this makes the sections easier to follow. Does this meet your expectations?
>
> **Analysis of overparameterization risk on the small dataset or Empirical results on larger datasets: The dataset is small. Any overparameterization risk? It means that the number of learnable parameters is much larger than the number of training samples. The authors should analyze the influence of this risk.**
>
> The first draft of our paper did not include a complete experiment description, and we have revised it accordingly. We use a large dataset of 304 games to pre-train the Siamese network, then perform the active learning study on a subset of them.
>
> We agree that more extensive experiments with more samples and study participants can be preferable in general. But we capped the dataset size of the user study because we consider one game to be a minimal but representative sample size. This way, we control the experiment's side effects and costs, so that we can see and compare the performance of the different methods.
>
> We have revised the description of the "dataset" (Section 3.4) accordingly.

---

### Decision · Action_Editors · 2023-03-14

**Recommendation:** Accept with minor revision

**Comment:**

This paper proposes an entropy-based active learning algorithm to do football trajectory matching. The reviewers agree that this is an interesting application with solid experimental results and a user study conducted by the authors. However, the common consensus is that the paper can be strengthened by showing the potential impact of the proposed method with more large-scale datasets and by adding more extensive baseline methods. There were several comments on the presentation of the paper, some of which have already been addressed. At this point, I recommend acceptance with a minor revision, where the authors address all presentational and clarification questions raised by the reviewers.

**Audience:**

Yes, I believe the application and the user study are of interest.

**Claims And Evidence:**

Yes, to the best of my knowledge, there are no concerns about unsupported claims.